# Revisiting the Link Between Keratoconus and Mitral Valve Prolapse

**Christian K. Five** [1,2], **Nina E. Hasselberg** [1,2], **Hilde Bjerkreim** [2,3], **Linda T. Aaserud** [1,2], **Anna Isotta Castrini** [1,2], **Cecilie Bugge** [1,2], **Eivind W. Aabel** [1,2], **Thomas Helle-Valle** [1], **Håvard Dalen** [1,4,5,6], **Olav Kristianslund** [2,3] and **Kristina H. Haugaa** [1,2,*]

[1]  ProCardio Center for Innovation, Department of Cardiology, Oslo University Hospital, Rikshospitalet, Nydalen, P.O. Box 4950, 0424 Oslo, Norway; five.christian@gmail.com (C.K.F.); ninaeha@hotmail.com (N.E.H.); l.s.a.t.aaserud@gmail.com (L.T.A.); isottacastrini@gmail.com (A.I.C.); cecilie.bugge@lyse.net (C.B.); eivind.westrum.aabel@gmail.com (E.W.A.); thelleva@ous-hf.no (T.H.-V.); havard.dalen@ntnu.no (H.D.)

[2]  Institute of Clinical Medicine, Faculty of Medicine, University of Oslo, 0372 Oslo, Norway; olav.kristianslund@gmail.com (O.K.)

[3]  Department of Ophthalmology, Oslo University Hospital, Ullevål, 0450 Oslo, Norway

[4]  Clinic of Cardiology, St. Olavs University Hospital, 7006 Trondheim, Norway

[5]  Clinic of Medicine, Levanger Hospital, 7601 Levanger, Norway

[6]  Department of Circulation and Medical Imaging, Norwegian University of Science and Technology, 7006 Trondheim, Norway

*  Correspondence: k.i.h.h.haugaa@medisin.uio.no

**Abstract:** Keratoconus is a progressive eye disease that results in thinning of the cornea, leading to visual impairment. Mitral valve prolapse (MVP) is a common disorder affecting around 2–4% of the general population. Previous studies have found an overrepresentation of MVP in individuals with keratoconus, with a prevalence of 38–65%, suggesting a shared underlying mechanism. In this case-control study, patients with keratoconus were enrolled from a quality and research registry. They were examined by a 2D echocardiography to identify if they had MVP, billowing or normal mitral leaflets. Controls were matched from the population-based Trøndelag Health Study. Patients and controls underwent a detailed echocardiographic examination to detect abnormal mitral valves. We included 101 patients (age 33 [IQR 29–40], 75% male) with keratoconus and 101 matched individuals. MVP was found in 2 (2%), while billowing was found in 5 (5%) of keratoconus patients. No significant association was found between keratoconus and the prevalence of MVP or billowing compared to the control group. Moreover, no associations were found between severity of keratoconus with presence of MVP nor with billowing of the mitral valves. We could not confirm the previously reported association between keratoconus and MVP, suggesting that routine screening for MVP in keratoconus patients may not be warranted. However, we cannot rule out the possibility of an association in other gender, age and ethnic groups different than ours.

**Keywords:** mitral valve prolapse; keratoconus; collagen defect

## 1. Introduction

Heart valves are composed of collagen types I and V and a smaller proportion of type III [1]. Mitral valve prolapse (MVP) is a common disorder affecting around 2–4% of the general population [2] and is more common in patients with systemic collagen disorders, like Marfan's syndrome and Ehlers–Danlos syndrome [3].

The cornea, like the heart valves, is composed of collagen. Degeneration of corneal collagen types I and V is hypothesized to be involved in the pathogenesis of keratoconus [4]. This is a progressive eye disease affecting approximately 0.1–0.2% of the population [5,6]. It is characterized by progressive corneal thinning, resulting in a conical shape of the cornea, and this can lead to vision loss if not properly diagnosed and treated. Keratoconus occurs at a young age (usually late teens), and there is currently no curative treatment. However, the treatment, corneal collagen cross-linking (CXL), has shown good results in halting the progression of the disease. CXL treatment is usually performed between the ages of 15 to 25 years, and seldom after the age of 30 years due to the natural crosslinking and stabilization of the cornea. This method aims to stabilize the progression of keratoconus by mechanical removal of the central corneal epithelium and crosslinking the collagen fibers by application of riboflavin and ultraviolet-A light treatment [7]. Like MVP, keratoconus has also been associated with systemic collagen disorders [4].

The similarities in collagen composition between MVP and keratoconus have previously led to a hypothesis that MVP and keratoconus may share similar pathophysiological etiologies, with collagen defects leading to structural changes in the mitral valve and the cornea [4]. Associations of keratoconus with MVP have been reported in several smaller studies [4,8–12], with an MVP prevalence ranging from 38 to 65% in persons with keratoconus [4,9–11]. A possible association has also been proposed between the severity of keratoconus and an increased incidence of MVP [11]. These reports, together with a large registry-based study [13], have suggested that patients with keratoconus should undergo echocardiographic screening for MVP, especially patients with the most severe forms of keratoconus [11,13]. However, this is not the clinical practice today. Furthermore, improvements in echocardiographic imaging resolution have enabled better visualization of cardiac valve, and the definitions of MVP have changed during the years [14,15]. How these changes affect the association between prevalence of keratoconus and MVP is unclear.

We aimed to explore the prevalence of mitral valve pathology, including mitral valve prolapse and billowing, in keratoconus patients treated with CXL using most modern echocardiographic technology and current recommendations for MVP diagnosis [15].

## 2. Materials and Methods

### 2.1. Study Population and Recruitment

Patients with a verified keratoconus diagnosis were enrolled in the study to undergo a cardiologic examination. These keratoconus patients were identified from a quality and research registry for CXL treatment and had previously been referred to the Department of Ophthalmology, Oslo University Hospital from January 2018 to June 2023, to confirm the diagnosis of keratoconus and evaluate the indication for CXL treatment. The diagnosis was established using Pentacam (Oculus) Scheimpflug tomography, which included measurement of maximum keratometry (K-max) in Diopter (D) and the corneal thickness together with signs of progression [16]. Patients with the following inclusion criteria were invited for an echocardiography as part of the present study: $\geq$18 years of age, living in the Oslo area, and a definite diagnosis of keratoconus.

Patients with known syndromic or connective tissue disease (Marfan's syndrome, Ehlers–Danlos syndrome, Osteogenesis imperfecta and Down syndrome) were excluded [9,11,12,17].

All patients underwent a cardiac evaluation including a clinical history, clinical cardiac examination, 12-lead electrocardiogram (ECG) and 2D transthoracic echocardiography. We assessed previous heart-related symptoms, chest pain, palpitations, dizziness, shortness of breath and syncope. We recorded co-morbidities and medications and family history. For comparison, we included a 1:1 age-, gender-, height-, and weight-matched control group

from the echocardiographic sub study (HUNT4Echo) in the fourth wave of the HUNT Study of individuals examined with echocardiography as part of their assessment in the population-based Trøndelag Health Study (HUNT), Norway [18]. Comprehensive details of the HUNT4Echo study have been previously published [19,20].

### 2.2. CXL Treatment Registry

Patients with keratoconus were recruited from the CXL treatment registry. This is a quality and research registry for all patients who have received CXL treatment for keratoconus in Norway. Patients fulfilling inclusion criteria were contacted by an invitation letter.

### 2.3. Echocardiography

All patients were examined using the GE Healthcare Vivid E95 cardiac ultrasound system with 2D echocardiography. Imaging data were analyzed offline (echocardiographic data by EchoPAC v203 [GE Healthcare, Horten, Norway].

We defined MVP as a superior displacement $\geq 2$ mm of any part of the mitral leaflet beyond the mitral annulus on echocardiography in the parasternal long-axis view [15]. Mitral valve billowing was defined as superior displacement between 0.1 and 1.9 mm of any part of the mitral leaflet beyond the mitral annulus. We graded mitral regurgitation (MR) according to current guidelines [21]. Left ventricular ejection fraction (EF) was assessed by Simpson's biplane method or auto EF. Cardiac volumes and function were assessed according to guidelines [22].

The analyses of the echocardiographic images were made by two observers (CKF and HD). Interobserver analyses on MVP and billowing diagnosis were performed in 30 keratoconus patients (CKF and THV) and in 30 control patients (CKF and HD) (Supplemental Table S1).

### 2.4. Eye Examinations

All keratoconus patients were examined with Pentacam (Oculus Optikeräte GmbH, Wetzlar, Germany) Scheimpflug tomography, uncorrected and corrected distance visual acuity, and slit lamp examination. The diagnosis of keratoconus was defined by the typical pattern observed on corneal tomography, particularly posterior surface elevation and increased K-max, along with clinical signs [16]. Indication for CXL treatment was based on assessed or suspected disease progression. We classified patients with moderate or severe keratoconus, with moderate defined as having K-max between 52 and 56 D and severe above 56 D [23].

### 2.5. Statistical Analyses

Values were expressed as mean $\pm$ standard deviation (SD), frequencies (%) or median with interquartile range (IQR). Groups were compared with independent Student's *t*-test, Mann–Whitney U test, chi squared or Fisher exact tests, as appropriate. Analyses were performed on separate groups of patients with MVP, billowing and the combined group defined as abnormal mitral valves (including both MVP and billowing), comparing them to the control group, as well as comparing patients with severe and moderate keratoconus. Two-sided *p*-values < 0.05 were considered significant (Stata/SE v18.0, StataCorp LLC, Texas, TX, USA)

## 3. Results

### 3.1. Study Population

Out of the 371 patients treated for keratoconus in the CXL registry, 225 patients fulfilled the inclusion criteria and were invited to participate in the study. Among these, 114 (51%)

patients responded, and ultimately 101 (45%) patients were able to attend the cardiac examinations (age 33 years [IQR 29 to 40], 75% male) (Figure 1 and Table 1). They had a mean K-max of 57.8 D (SD 6.7) and a minimal thickness of the cornea of 460 μm (SD 40). Mean age at CXL treatment was 30 years (IQR 26–37). Regarding cardiac symptoms, 14 (14%) had a history of chest pain, 19 (19%) of palpitations and 7 (7%) of syncope. The most common comorbidities were seven (7%) with hypertension, five (5%) with asthma, and four (4%) had rheumatic diseases. Four (4%) had family members with keratoconus, and 11 (11%) had family members with a history of cardiac disease (Table 1).

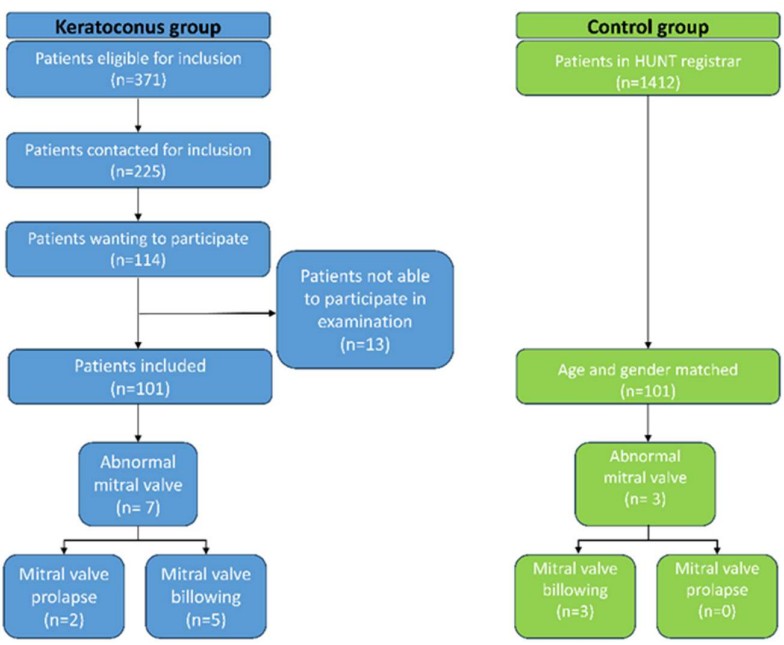

**Figure 1.** Study population, recruitment, and mitral valve prolapse or billowing.

**Table 1.** Clinical characteristics and echocardiographic parameters of the 101 keratoconus and 101 age- and gender-matched control subjects.

|  | **Total** **n = 202** | **Patients with Keratoconus** **n = 101** | **Control Patients** **n = 101** | ***p =*** |
|---|---|---|---|---|
| Age at inclusion, year (IQR) | 35 (30–41) | 33 (29–40) | 36 (32–41) | 0.18 |
| Male, n (%) | 152 (75) | 76 (75) | 76 (75) | 1.00 |
| Height, cm (SD) | 178 (9) | 177 (10) | 179 (9) | 0.41 |
| Weight, kg (SD) | 88 (18) | 88 (20) | 87 (16) | 0.65 |
| BMI, (SD) | 28 (6) | 28 (6) | 27 (5) | 0.29 |
| BSA, (SD) | 2.07 (0.23) | 2.08 (0.25) | 2.05 (0.20) | 0.40 |
| Heart rate, BPM (SD) | 70 (14) | 72 (16) | 68 (11) | 0.05 |
| Systolic BP, mmHg (SD) | 126 (13) | 130 (13) | 121 (11) | <0.01 |
| Diastolic BP, mmHg (SD) | 74 (9) | 76 (9) | 73 (8) | 0.02 |
| Heart murmur, n (%) |  | 5 (5) | n.a. |  |
| Comorbidities |  |  |  |  |
| Hypertension, n (%) | 7 (3) | 7 (7) | 0 (0) | 0.01 |
| Asthma, n (%) | 16 (8) | 5 (5) | 11 (11) | 0.11 |
| Rheumatic disease, n (%) | 5 (3) | 4 (4) | 1 (1) | 0.37 |
| Symptoms |  |  |  |  |
| Chest pain, n (%) | 21 (10) | 14 (14) | 7 (7) | 0.10 |
| Palpitation, n (%) | 31 (15) | 19 (19) | 12 (12) | 0.19 |
| Syncope, n (%) | 10 (5) | 7 (7) | 3 (3) | 0.33 |

**Table 1.** *Cont.*

| | Total n = 202 | Patients with Keratoconus n = 101 | Control Patients n = 101 | p = |
|---|---|---|---|---|
| Family history | | | | |
| Family history of cardiac disease | 22 (11) | 11 (11) | 11 (11) | 1.00 |
| Family history of keratoconus | | 4 (4) | n.a. | |
| Echocardiogram | | | | |
| Abnormal mitral valve, n (%) | 10 (5) | 7 (7) | 3 (3) | 0.33 |
| Mitral valve prolapse, n (%) | 2 (1) | 2 (2) | 0 (0) | 0.50 |
| Mitral valve billowing, n (%) | 8 (4) | 5 (5) | 3 (3) | 0.72 |
| Mitral insufficiency, n (%) | 33 (16) | 14 (14) | 19 (19) | 0.34 |
| Grade | | | | |
| 1 | 31 (16) | 14 (14) | 17 (17) | 0.52 |
| 2 | 2 (1) | 0 (0) | 2 (2) | 0.15 |
| 3 | 0 (0) | na | na | na |
| Interventricular septum diameter, mm (SD) | 7.9 (1.6) | 7.9 (1.8) | 8.0 (1.5) | 0.61 |
| Left ventricle end diastole diameter, mm (SD) | 51.8 (4.8) | 51.4 (5.0) | 52.2 (4.7) | 0.24 |
| Ejection fraction, % (SD) | 59 (6) | 58 (6) | 59 (5) | 0.57 |

*p* values from Student's *t*-test, Mann–Whitney U test, chi-square test or Fisher exact test, as appropriate. BMI = body mass index; BP = blood pressure; BPM = beats per minute; BSA = body surface area; IQR = interquartile range; SD = standard deviation.

From the 371 patients in the registry for CXL-treated keratoconus patients, a total of 225 patients were invited to participate in this study. A total of 114 patients responded in favor of being included in the study, and a total of 101 patients managed to partake in the cardiac evaluation. Of the 101 patients, we found seven patients to have an abnormal mitral valve. Of these seven, two had mitral valve prolapse, and five patients had billowing. We used the region-based HUNT registry to find 101 age- and gender-matched controls. Here, we found three patients with abnormal mitral valve, no cases of mitral valve prolapse, and three cases of billowing. CXL = corneal collagen cross linking, HUNT = Trøndelag Health Study.

*3.2. Echocardiographic Findings in Patients with Keratoconus Compared to the Control Group*

Among the 101 patients with keratoconus, two (2%) patients had MVP compared to zero (0%) patients in the control group (*p* = 0.50) (Table 1). Furthermore, five (5%) keratoconus patients had mitral valve billowing, compared to three (3%) control patients (*p* = 0.72). Furthermore, there was no difference between the keratoconus and control patients regarding the presence or severity of mitral insufficiency (*p* = 0.34 and *p* = 0.15) (Table 1), nor in left ventricular end-diastolic diameter (*p* = 0.24) or ejection fraction (*p* = 0.57).

*3.3. Characteristics of Keratoconus Patients with MVP or Mitral Valve Billowing*

Patient #1 with MVP had posterior leaflet prolapse and billowing of the anterior leaflet. CXL treatment was performed on one eye at age 31 years with K-Max of 49.2 D, and the thinnest corneal thickness was 513 μm. The patient had a history of chest pain and syncope and a family history of sudden cardiac death, leading to further cardiac follow-up. Comorbidities included pericarditis and previous treatment with prednisolone. Patient #2 with MVP had isolated anterior leaflet prolapse and mitral annular disjunction and a

history of chest pain. This patient received CXL treatment at age 26 years with K-max of 53.3 D, and the thinnest corneal thickness was 465 μm.

K-Max and corneal thickness in patients #1 and #2 did not differ compared to the rest of the keratoconus patients. ($p$ = 0.24 and $p$ = 0.31, respectively).

Additionally, we identified five (5%) patients with billowing of either leaflet, summing to a total of seven (7%) keratoconus patients with an abnormal mitral valve. We found no differences between keratoconus patients with abnormal or normal mitral valves regarding age at echocardiographic examination (34 years [IQR 28–35] vs. 33 years [IQR 29–41] $p$ = 0.67), age at CXL treatment (31 years [IQR 26–31] vs. 30 years [IQR 26–37] $p$ = 0.88), K-max values (54.6 D [SD 4.4] vs. 57.0 D [SD 6.9] $p$ = 0.37), or minimal thickness of cornea (463 μm [SD 28] vs. 460 μm [SD 41] $p$ = 0.83). None of the 7 patients with abnormal mitral valves had performed bilateral CXL treatment, compared to 32 (34%) of the keratoconus patients with normal mitral valves, ($p$ = 0.09). (Table 2). We found no difference in the occurrence of comorbidities between the groups, including asthma ($p$ = 1.00), hypertension ($p$ = 1.00), rheumatic disease ($p$ = 0.25) or reported symptoms (Table 2).

**Table 2.** Clinical characteristics of the 101 keratoconus patients, divided by normal mitral valve and abnormal mitral valve.

| | Total n = 101 | Normal Mitral Valve n = 94 | MVP or Billowing n = 7 | $p$ = |
|---|---|---|---|---|
| Age at inclusion, year (IQR) | 33 (29–40) | 33 (29–41) | 34 (28–35) | 0.67 |
| Male, n (%) | 76 (75) | 71 (76) | 5 (71) | 1.00 |
| Height, cm (SD) | 178 (10) | 177 (10) | 180 (11) | 0.47 |
| Weight, kg (SD) | 89 (20) | 90 (20) | 80 (13) | 0.24 |
| BSA, (SD) | 2.1 (0.2) | 2.1 (0.3) | 2.0 (0.1) | 0.33 |
| Systolic BP, mmHg (SD) | 130 (13) | 131 (12) | 124 (16) | 0.15 |
| Diastolic BP, mmHg (SD) | 76 (9) | 76 (9) | 74 (8) | 0.61 |
| Heart murmur, n (%) | 5 (5) | 5 (6) | 0 (0) | 1.00 |
| Symptoms | | | | |
| Chest pain, n (%) | 14 (14) | 12 (13) | 2 (29) | 0.25 |
| Palpitation, n (%) | 18 (18) | 15 (16) | 3 (43) | 0.11 |
| Syncope, n (%) | 7 (7) | 5 (5) | 2 (29) | 0.07 |
| Family history | | | | |
| Family history of keratoconus, n (%) | 4 (4) | 3 (3) | 1 (14) | 0.25 |
| Family history of cardiac disease, n (%) | 11 (11) | 10 (11) | 1 (14) | 0.56 |
| Comorbidities | | | | |
| Hypertension, n (%) | 7 (7) | 7 (7) | 0 (0) | 1.00 |
| Asthma, n (%) | 5 (5) | 5 (5) | 0 (0) | 1.00 |
| Rheumatic disease, n (%) | 4 (4) | 3 (3) | 1 (14) | 0.25 |
| Echocardiogram | | | | |
| Mitral insufficiency, n (%) | 14 (14) | 12 (13) | 2 (29) | 0.25 |
| Inter ventricular septum, mm (SD) | 7.9 (1.8) | 7.9 (1.8) | 7.4 (1.9) | 0.48 |
| Left ventricle end diastole diameter, mm (SD) | 51.4 (4.9) | 51.3 (5.1) | 53.3 (2.8) | 0.31 |
| Ejection fraction, % (SD) | 58 (6) | 59 (6) | 57 (6) | 0.46 |

**Table 2.** *Cont.*

| | Total n = 101 | Normal Mitral Valve n = 94 | MVP or Billowing n = 7 | *p* = |
|---|---|---|---|---|
| ECG | | | | |
| Atrial fibrillation, n (%) | 0 (0) | 0 (0) | 0 (0) | n.a |
| T-inversion in 1 lead, n (%) | 10 (10) | 10 (11) | 0 (0) | 1.00 |
| T-inversion in 2 or more leads, n (%) | 1 (1) | 1 (1) | 0 (0) | 1.00 |
| Incomplete RBB | 8 (8) | 8 (8) | 0 (0) | 1.00 |
| AV-block grad 1 | 3 (3) | 3 (3) | 0 (0) | 1.00 |
| Eye examinations | | | | |
| K-max, D (SD) | 57.8 (6.7) | 57.0 (6.9) | 54.6 (4.4) | 0.37 |
| Minimal thickness of cornea, μm (SD) | 460 (40) | 460 (41) | 463 (28) | 0.83 |
| Bi-lateral CXL treatment, n (%) | 32 (32) | 32 (34) | 0 (0) | 0.09 |
| Age at CXL treatment, years (IQR) | 30 (26–37) | 30 (26–37) | 31 (26–31) | 0.88 |
| Severe keratoconus, n (%) | 51 (50) | 48 (51) | 3 (43) | 0.71 |

*p* values from Student's *t*-test, Mann–Whitney U test, chi-square test or Fisher exact test, as appropriate. AV-block = atrio ventricular block; BP = blood pressure; BSA = body surface area; CXL = corneal collagen cross linking; D = Dioptres; ECG = electrocardiogram; IQR = interquartile range; K-max = maximum keratometry; RBB = right bundle branch block; SD = standard deviation.

### 3.4. Abnormal Mitral Valve in Patients with Severe Compared to Moderate Keratoconus

Fifty patients (50%) had moderate keratoconus, and 51 (50%) had severe keratoconus (K-max cut-off value 56 D) [23]. There was no difference between patients with moderate and severe keratoconus regarding age at echocardiographic examination ($p = 0.22$) or in the proportion of males ($p = 0.45$). Those with severe keratoconus had borderline higher frequency of asthma (five [10%] patients vs. zero [0%] patients, $p = 0.06$), and had thinner corneas compared to those with moderate keratoconus (442 μm [SD 36] vs. 478 μm [SD 36], $p < 0.01$) (Table 3).

**Table 3.** Clinical characteristics of the 101 keratoconus patients, divided into moderate (K-max < 56) and severe (K-max > 56) keratoconus.

| | Total n = 101 | Moderate Keratoconus n = 50 | Severe Keratoconus n = 51 | *p* = |
|---|---|---|---|---|
| Age at inclusion, year (IQR) | 33 (29–40) | 34 (30–42) | 33 (29–38) | 0.22 |
| Male, n (%) | 76 (75) | 36 (72) | 40 (78) | 0.45 |
| Height, cm (SD) | 178 (10) | 176 (9) | 179 (11) | 0.25 |
| Weight, kg (SD) | 89 (20) | 85 (13) | 92 (24) | 0.09 |
| BMI, (SD) | 28 (6) | 27 (4) | 29 (8) | 0.24 |
| BSA, (SD) | 2.08 (0.25) | 2.04 (0.18) | 2.12 (0.29) | 0.10 |
| Systolic BP, mmHg (SD) | 130 (13) | 129 (10) | 132 (15) | 0.36 |
| Diastolic BP, mmHg (SD) | 76 (9) | 74 (9) | 77 (8) | 0.09 |
| Heart rate, beats per minute (SD) | 72 (16) | 72 (17) | 71 (14) | 0.66 |
| Heart murmur, n (%) | 5 (5) | 4 (8) | 1 (2) | 0.16 |
| Symptoms | | | | |
| Chest pain, n (%) | 14 (14) | 10 (20) | 4 (8) | 0.09 |
| Palpitation, n (%) | 18 (18) | 10 (20) | 8 (16) | 0.57 |
| Syncope, n (%) | 7 (7) | 3 (6) | 4 (8) | 1.00 |

**Table 3.** *Cont.*

| | Total n = 101 | Moderate Keratoconus n = 50 | Severe Keratoconus n = 51 | *p* = |
|---|---|---|---|---|
| **Family history** | | | | |
| Family history of keratoconus, n (%) | 4 (4) | 2 (4) | 2 (4) | 1.00 |
| Family history of cardiac disease, n (%) | 11 (11) | 3 (6) | 8 (16) | 0.20 |
| **Comorbidities** | | | | |
| Hypertension, n (%) | 7 (7) | 2 (4) | 5 (10) | 0.44 |
| Asthma, n (%) | 5 (5) | 0 (0) | 5 (10) | 0.06 |
| Rheumatic disease, n (%) | 4 (4) | 2 (2) | 2 (4) | 1.00 |
| **Echocardiogram** | | | | |
| Abnormal mitral valve, n (%) | 7 (7) | 4 (8) | 3 (6) | 0.70 |
| Mitral valve prolapse, n (%) | 2 (2) | 2 (4) | 0 (0) | 0.24 |
| Mitral valve billowing, n (%) | 5 (5) | 2 (4) | 3 (6) | 1.00 |
| Mitral insufficiency, n (%) | 14 (14) | 10 (20) | 4 (8) | 0.09 |
| Inter ventricular septum, mm (SD) | 7.8 (1.8) | 7.7 (1.5) | 8.0 (2.0) | 0.42 |
| Left ventricle end diastole diameter, mm (SD) | 51.4 (5.0) | 51.5 (5.0) | 51.4 (5.0) | 0.89 |
| Ejection fraction, % (SD) | 58 (6) | 58 (5) | 59 (7) | 0.50 |
| **ECG** | | | | |
| Atrial fibrillation, n (%) | 0 (0) | 0 (0) | 0 (0) | na |
| T-inversion in 1 lead, n (%) | 10 (10) | 7 (14) | 3 (6) | 0.20 |
| T-inversion in 2 or more leads, n (%) | 1 (1) | 0 (0) | 1 (2) | 1.00 |
| Incomplete RBB, n (%) | 8 (8) | 4 (8) | 4 (8) | 1.00 |
| First-degree AV-block, n (%) | 3 (3) | 1 (2) | 2 (4) | 1.00 |
| **Eye examinations** | | | | |
| Minimal thickness of cornea, μm (SD) | 460 (40) | 478 (36) | 442 (36) | <0.01 |
| Bi-lateral CXL treatment, n (%) | 32 (32) | 17 (34) | 15 (30) | 0.67 |
| Age at CXL treatment, years (IQR) | 30 (26–37) | 31 (27–37) | 30 (26–35) | 0.50 |

*p* values from Student's *t*-test, Mann–Whitney U test, chi-square test or Fisher exact test, as appropriate. AV-block = atrio ventricular block; BMI = body mass index; BP = blood pressure; BSA = body surface area; CXL = corneal collagen cross linking; ECG = electrocardiogram; IQR = interquartile range; K-max = maximum keratometry; RBB = right bundle branch block; SD = standard deviation.

We found no difference in occurrence of MVP ($p = 0.24$), billowing ($p = 1.00$) or abnormal mitral valve ($p = 0.70$) nor in any ECG changes between the groups (Table 3).

## 4. Discussion

This study did not find an overrepresentation of abnormal mitral valves, including MVP or billowing in patients with keratoconus compared to a normal population. Furthermore, MVP was not more prevalent in those with the most severe compared to moderate keratoconus. Thus, the results from our study do not support screening of all patients with keratoconus for abnormal mitral valves in patients with keratoconus.

### 4.1. Relationship Between Keratoconus and MVP Prevalence

Previous studies have shown diverging results regarding the potential association between keratoconus and MVP [8–12,17]. We found a prevalence of MVP of 2% and of

mitral valve abnormalities of 7%. These prevalences were similar in our 101 patients with keratoconus and in 101 matched healthy individuals recruited from the population-based HUNT study. The prevalence of MVP among the keratoconus patients was consistent with previously reported MVP prevalence of 2 to 3% in the general population [2,24]. These findings were in line with a previous study of 95 patients with keratoconus which found no overrepresentation of MVP among the keratoconus group [17].

In contrast, other previous studies found a prevalence of MVP ranging between 38 and 65% among keratoconus patients, which was significantly higher than the prevalence of MVP in their control groups ranging between 7 and 13% [9–11].

The possible explanation for the discrepancies in MVP prevalence is that previous studies defined MVP according to different diagnostic criteria and methods. One study from 2011 [11] used the Perloff criteria for MVP definition [14]. These criteria use the echocardiographic apical four chamber view for visualizing MVP, instead of the parasternal long axis view which is current practice. In addition, patients were diagnosed with MVP when auscultation revealed mild to late systolic click and late systolic murmur over the apex, even in the absence of any echocardiographic findings [11]. Other studies did not specify the echocardiographic view or method used in the diagnosis of MVP [9,10]. Later studies have shown that using the echocardiographic apical four-chamber view for diagnosing MVP, increases the number of MVP diagnoses [25,26]. This could explain the higher frequency of MVP reported in these previous studies, both among keratoconus patients and controls [9–11] compared to our study. However, the use of different echocardiographic views and diagnostic criteria cannot explain the higher prevalence of MVP in patients with keratoconus compared to controls observed in these early studies [9–11].

A previous study indicated a progression from mitral valve billowing to later development of MVP [27]. Therefore, we combined groups of mitral valve billowing and MVP to ensure that we did not miss a possible association between early-stage MVP and keratoconus. However, even including these patients did not yield a significant association between abnormal mitral valves and keratoconus, in contrast to previous studies [8–12].

Another possible explanation of our low MVP prevalence is that our cohort mostly consisted of younger male patients. The population study from Taiwan [13] showed that the subgroup of women above 40 years of age had significantly higher MVP prevalence, and this group was the determining factor for the significantly higher prevalence among keratoconus patients. Our cohort included only three female patients above 40 years of age. Male gender has previously been reported to be overrepresented in the keratoconus population [28]. However, larger population-based studies on MVP prevalence have not found differences between the genders or between age groups [2]. Our study cannot exclude that women above 40 years of age with keratoconus have a higher prevalence of MVP and that there may be differences among ethnicities.

### 4.2. Severity of Keratoconus and Cardiac Findings

We did not find a higher prevalence of MVP in those with the more severe form of keratoconus, in contrast to a previous study [11]. Rabbanikhah et al. defined severe keratoconus as having acute corneal hydrops [11]. This is not directly comparable to using K-Max for grading keratoconus severity as defined in our study, and our cohorts may therefore differ substantially in keratoconus severity. All our patients were treated with CXL, and it is unknown how many of our patients would develop acute corneal hydrops if left untreated. Furthermore, by using the CXL registry for inclusion, patients with the most severe keratoconus may have had contraindications for CXL treatment, and therefore not been included in the CXL registry and our study. And we may have missed an association due to this.

We found a borderline significant association between severe keratoconus and the prevalence of asthma, as previously suggested [29]. We also found an association between severe keratoconus and a decreased corneal thickness, indicating the relevance of K-max as a marker for keratoconus severity.

### 4.3. Clinical Implications

Our findings did not support a general recommendation to screen keratoconus patients for MVP. However, our keratoconus patient demographics were dominated by young men, and we cannot exclude that women above 40 of age with keratoconus may benefit for MVP screening due to previous findings in this group [13].

### 4.4. Limitations

This was a single-center study with its inherent limitations. The sample size was limited, although it is the largest echocardiographic study reported in patients with keratoconus. We included patients treated with CXL and may, therefore, have missed patients with most severe forms of keratoconus who were deemed not eligible for CXL treatment. The use of K-Max and billowing has its limitations regarding reproducibility.

The cohort consisted predominantly of young male patients of Norwegian ethnicity. The impact of ethnicity and environmental factors on MVP are not known and may be other potential explanations for the difference in MVP prevalence in our study compared to previous studies performed in Asian countries [11,13] and in the United Kingdom [9].

## 5. Conclusions

This study showed no clear association between keratoconus and the prevalence of MVP or mitral valve billowing in a cohort composed of predominantly young males. Our findings did not support MVP screening of all patients with keratoconus. However, we cannot exclude an association between MVP and keratoconus in other gender, age and ethnic groups.

**Supplementary Materials:** The following supporting information can be downloaded at: https://www.mdpi.com/article/10.3390/cardiogenetics15010004/s1, Table S1: Interobserver analyses on MVP and billowing.

**Author Contributions:** Conceptualization, N.E.H., H.B., E.W.A., O.K. and K.H.H.; methodology, C.K.F., N.E.H., O.K., K.H.H. and A.I.C.; software; validation T.H.-V. and H.D.; formal analysis, C.K.F., N.E.H. and H.B. investigation, C.K.F., T.H.-V., H.D. and O.K.; resources, O.K. and K.H.H.; data curation, C.K.F., E.W.A. and K.H.H.; writing—original draft preparation, C.K.F.; writing—review and editing, C.K.F., N.E.H., H.B., L.T.A., A.I.C., C.B., E.W.A., T.H.-V., H.D., O.K. and K.H.H.; visualization, C.K.F.; supervision, O.K. and K.H.H.; project administration, C.K.F. and N.E.H.; funding acquisition, O.K. and K.H.H. All authors have read and agreed to the published version of the manuscript.

**Funding:** This work was funded by the Norwegian Research council, ProCardio (grant number 309762) to CKF, KH, NEH, CB, LTA), GENE POSITIVE (grant number 288438) to AIC, EWA.

**Institutional Review Board Statement:** This study was conducted in accordance with the Declaration of Helsinki and approved by the Regional Committee for Medical Research Ethics (283731/REK Nord).

**Informed Consent Statement:** Informed consent was obtained from all subjects involved in the study.

**Data Availability Statement:** The data underlying this article cannot be shared publicly due to the privacy of individuals that participated in the study. The data will be shared on reasonable request to the corresponding author.

**Acknowledgments:** The Trøndelag Health Study (HUNT) is a collaboration between HUNT Research Centre (Faculty of Medicine and Health Sciences, Norwegian University of Science and Technology NTNU), Trøndelag County Council, Central Norway Regional Health Authority, and the Norwegian Institute of Public Health. We would also like to thank the patients for participating in this study.

**Conflicts of Interest:** The authors have no conflicts of interest to disclose.

## Abbreviations

| | |
|---|---|
| CXL | corneal cross-linking |
| CKF | Christian Kullmann Five |
| ECG | electrocardiogram |
| EF | ejection fraction |
| HD | Håvard Dalen |
| HUNT | Trøndelag Health Study |
| IQR | interquartile range |
| K-max | maximum keratometry |
| MR | mitral regurgitation |
| MVP | mitral valve prolapse |
| SD | standard deviation |
| THV | Thomas Helle-Valle |

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
