# Peer review of "Revisiting the Link Between Keratoconus and Mitral Valve Prolapse"

_cardiogenetics, doi:10.3390/cardiogenetics15010004_

Round 1
Reviewer 1 Report
Comments and Suggestions for Authors
This manuscript is well-written and presents potentially clinically significant findings. Although the study is relatively large within the context of keratoconus research, the data remain limited, which reduces its overall persuasiveness. Notably, among familial cases, the proposed genetic causes of keratoconus (e.g., VSX1—visual system homeobox 1) differ from those of MVP (e.g., DCHS1—dachsous cadherin-related 1). Hence, it is not surprising that the findings reported by Dr. Five et al. do not support a link between these two conditions.
Reviewer 2 Report
Comments and Suggestions for Authors
The article by Five et al. argues that the reported link between keratoconus and mitral valve prolapse is likely a false positive. I fully agree with this conclusion, as I have not found any studies using modern methods, such as high-throughput sequencing, that support this connection.
For example, Karolak et al. (2017) identified genetic variants causing keratoconus but did not find any link to mitral valve prolapse. The idea of a connection between these two conditions seems to come from the meta-analysis by Siordia JA and Franco JC (2020), which used clinical reports and did not include genetic evidence.
In contrast, Five et al. provide strong clinical data from over 300 patients, making their study reliable and well-supported.
I hope that you will have the opportunity to do whole genome or whole exome sequencing of the collected group of patients in your future studies and find the genetic causes of both diseases.
Reviewer 3 Report
Comments and Suggestions for Authors
Dear authors, I was reviewing with interest the manuscript entitled "Revisiting the link between keratoconus and mitral valve prolapse". The study is well done and is the largest echocardiographic study reported in patients with keratoconus. The criteria for the diagnosis of MVP is exactly defined and explains the differences, to previously published data. The results are straight and clear presented, the literature is appropriate and actual. In my opinion, the manuscript is worth being published in Cardiogenetics.